# OpenReview forum: "Can Transformers Learn Full Bayesian Inference in Context?"
_ICML.cc/2025/Conference — ICML 2025 poster_

### Official Review · Reviewer_qsBb · 2025-03-01

**Overall Recommendation:** 3

**Summary:**

The paper introduces an innovative approach to full Bayesian inference using in-context learning (ICL) with transformers. By leveraging ideas from continuous normalizing flows and flow matching, the authors propose a framework that learns to approximate the posterior distribution P(z|x) directly from synthetic samples drawn from the joint distribution. Evaluations on generalized linear models (GLMs), factor analysis, and Gaussian mixture models demonstrate that the ICL method produces posterior samples comparable to those obtained via Hamiltonian Monte Carlo and state‐of‐the‐art variational inference techniques. The work is well-motivated and offers a promising new direction by uniting meta-learning ideas with Bayesian inference.

**update after rebuttal**

I thank the authors for their responses. Although it is understandable that not many new empirical evidence can be provided due to the restricted length of rebuttal this year, the empirical evaluation remains the weakness of this paper. Hence I will maintain my current score. I disagree that image datasets are less relevant, since there are many works (e.g, Bayesian neural networks, and VAE related works) trying expand Bayesian inference to the image domain. I think potential evaluations on image datasets (with Bayesian versions of vision transformers) would greatly strengthen the paper in the future.

**Claims And Evidence:**

The central claim that transformers can learn full Bayesian inference in context is well-supported by extensive experiments on GLMs, factor analysis, and GMMs—showing that the posterior samples produced by the ICL approach closely match those obtained via HMC and often outperform several VI methods on synthetic and limited real-world datasets.

However, the claims on large-scale training is not well-backed by experiments. Although real-world data are used, they remain at tabular datasets. More advanced and large datasets like image datasets should be used to verify the effectiveness of the proposed frameworks. Also, some image generation tasks (e.g., on MNIST) can be included to test the quality of the learned posterior.

Somehow, what the authors claim is related to density estimation methods. Experiments on UCI datasets and related density estimation methods would better support the claims.

**Essential References Not Discussed:**

A comparison to related density estimation methods is missing, e.g., Salazar, Sebastian. "VaRT: variational regression trees." Advances in Neural Information Processing Systems 36 (2023): 45681-45693.

**Experimental Designs Or Analyses:**

The experiments and designs are generally adequate and sound.

**Methods And Evaluation Criteria:**

The proposed methods and evaluation criteria generally make sense for the problem at hand. The authors introduce a transformer-based in-context learning approach that leverages continuous normalizing flows and flow matching to approximate full posterior distributions. This method is evaluated using well-established metrics—C2ST, MMD, and Wasserstein—which are standard for comparing how close the inferred distributions are to those obtained by gold-standard methods like HMC. Furthermore, the experiments span a range of models (GLMs, factor analysis, and GMMs) and use both synthetic and curated real-world tabular datasets, which is appropriate for demonstrating feasibility across different Bayesian inference scenarios.

However, while the chosen benchmarks and evaluation criteria are solid for controlled experiments, the reliance on synthetic data and moderate-scale real-world datasets raises questions about the scalability and robustness of the method when dealing with more complex, high-dimensional tasks. And the baselines compared are insufficient - the authors only verify the comparability to existing optimization methods, but yet its practicability to different downstream tasks. Somehow what the authors are claiming are related to density estimation methods. Experiments on UCI datasets and related density estimation methods would better support the claims.

**Other Comments Or Suggestions:**

N.A.

**Other Strengths And Weaknesses:**

Strength
* Novelty: The paper is novel. applies transformer-based in-context learning to perform full Bayesian inference—a departure from conventional MCMC or VI methods.
* Flexibility: The proposed approach can handle multivariate and complex posterior distributions, showing robust performance even in cases with non-standard posterior shapes (e.g., skewed or multi-modal distributions).
* Empirical Results: Comparative evaluations indicate that the ICL approach outperforms traditional VI methods in capturing the full posterior, particularly in synthetic experiments.

Weaknesses

* My largest concern is the scalability issues. The authors only adopt small datasets (e.g., simulated and tabular datasets by Grinsztajn metal.). Authors should test the capability of the proposed transformer on image datasets (e.g., MNIST or ImageNet) to verify their claims on scalability.

*Although the comprehensiveness of evaluation in the suggested settings, I think comparisons to density estimation methods (e.g., UCI) and image generation (for validating the quality of the posterior distribution) would be necessary. The current empirical evaluation is less sufficient.

**Questions For Authors:**

In practical settings where model misspecification is a concern, how robust is the ICL approach? Have the authors explored scenarios where the generative model does not perfectly capture the true data distribution?

**Relation To Broader Scientific Literature:**

As mentioned before, I think the task that the authors are more like posterior density estimation. In this sense, including comparisons to density estimation benchmark would be necessary (e.g., UCI) and even image generation tasks for higher dimensional cases.

**Theoretical Claims:**

Many theoretical results are only outlined rather than given in full detail. The results heavily follow previous work which are well-verified. Perhaps the authors need to include the key assumptions by these works here for better clarity.

---

> ### Author Rebuttal · Authors · 2025-03-30
>
> Thank you for taking the time to read our manuscript and for providing detailed feedback.
>
> >More advanced and large datasets like image datasets should be used to verify the effectiveness of the proposed frameworks.
>
> Please note that, to the best of our knowledge, our paper presents the first thorough investigation of in-context learning for full Bayesian inference. We think that tabular data is a very natural domain for full Bayesian inference and in particular for the latent variable models considered in our work. While fully Bayesian methods for image data exist, they are arguably substantially less prominent. Furthermore, we believe that the inherent heterogeneity of the 17 considered real-world tabular datasets, spanning domains such as superconductors, environmental pollution, or wine quality, is well suited to provide a challenging and diverse benchmark.
>
> Following your comment, we now also include an [ablation study](https://anonymous.4open.science/r/Extra-B820/Dim_Ablation.pdf) that investigates the effect of the dimensionality of the used data.
>
> > Also, some image generation tasks (e.g., on MNIST) can be included to test the quality of the learned posterior.
>
> We would like to argue that image generation is usually not considered to be within the scope of full Bayesian inference, which we investigate in this paper.
>
> > Somehow, what the authors claim is related to density estimation methods. Experiments on UCI datasets and related density estimation methods would better support the claims.
>
> We would like to emphasize that in this work, we consider Bayesian inference as the task of providing samples from the posterior. This is in scope of what MCMC methods do and thus also importantly allows to compare our sampling-based method and the VI baselines to HMC as a gold standard. Even though Bayesian inference can also be accomplished via density estimation, we would argue that this is a different task.
>
> > the authors only verify the comparability to existing optimization methods, but yet its practicability to different downstream tasks.
>
> We thank the reviewer for this comment. Exclusively comparing posterior samples themselves is, however, a common practice in amortized (simulation-based) inference [1]. We would also like to highlight that, in practical applications, the results of the latent variable models we study are arguably mostly analyzed in isolation rather than being used directly for downstream tasks, which motivates our choice of evaluation procedure.
>
> However, following your recommendation, we now include results for [new experiments](https://anonymous.4open.science/r/Extra-B820/Pred_Performance.pdf) where we use the posterior samples for the GLMs case for the purpose of prediction.
>
> > Perhaps the authors need to include the key assumptions by these works here for better clarity.
>
> Thank you for this point. We will include the assumptions underlying flow matching in our paper besides just referring to the literature.
>
> > A comparison to related density estimation methods is missing, e.g., Salazar, Sebastian. "VaRT: variational regression trees." Advances in Neural Information Processing Systems 36 (2023): 45681-45693.
>
> We will add a more extensive discussion of density estimation methods, including the aforementioned paper, to the revised version of the manuscript and how they relate to our approach.
>
> > Have the authors explored scenarios where the generative model does not perfectly capture the true data distribution?
>
> Please refer to Appendix H for experimental results regarding the OOD performance of our approach and to Appendix B for a more detailed discussion on model misspecification. We will expand our explanations of the situation regarding OOD performance in a new section in the revised version of the manuscript. We created a [new plot](https://anonymous.4open.science/r/Extra-B820/plot_ood.pdf) for this purpose.
>
> ## References
>
> [1] Lueckmann, Jan-Matthis, et al. "Benchmarking simulation-based inference." AISTATS 2021.

---

### Official Review · Reviewer_RMVg · 2025-03-10

**Overall Recommendation:** 3

**Summary:**

The paper proposes an in-context learning (ICL) approach for performing Bayesian inference over three classes of models: Generalized Linear Models (GLMs), Gaussian Mixture Models, and Factor Analysis (FA). The paper shows that their ICL method can produce similar posterior samples to Hamiltonian Monte Carlo (HMC).

**Claims And Evidence:**

* Claim 1: ICL yields posterior samples that are very similar to HMC:
    * This does seem to be the case from the experimental results. Looking specifically at distributional metrics for measuring similarity between samples.
* Claim 2: ICL samples are preferred over popular VI techniques across the experiments:
	* Default hyperparameters are used for the VI approaches according to the appendix (no hyperparameter optimization) and the VI approaches do not seem to be state-of-the-art in VI.

**Essential References Not Discussed:**

* Not that I am aware.

**Experimental Designs Or Analyses:**

* See related comments in strengths and weaknesses.

**Methods And Evaluation Criteria:**

* The benchmarks and evaluation criteria make sense.

**Other Comments Or Suggestions:**

* I could not find details of the size of the data trained on for each class of model. (If it is hiding in the appendix, I apologise for missing it.)
* The experiments on the OOD performance in the appendix are interesting and probably should be promoted to the main paper. I would suggest thinking of a new way of presenting these experiments though. Perhaps as a plot where the y-axis is the metric, and the x-axis is the KL-divergence between the training and test data distributions. (This is just a suggestion.)

**Other Strengths And Weaknesses:**

### Strengths:
* The paper is well-written and the presentation (Figure 1 etc.) is strong.
* The results seem to support the claims.
### Weaknesses
* The key weakness that is preventing me from providing a higher score is in the identification of the novelty of the work in comparison to Müller et al. I think the main difference (novelty) is in the architecture (Figure 2). The loss function appears to be comparable to Müller et al. As such, it would seem like comparing to prior-data fitted networks from that paper, would really help show if the proposed ICL approach is superior.
* Another component of the Müller et al. paper that is missing from this paper is a reference to how many synthetic datasets were needed to achieve the reported performances. For example, it would be good to know how many synthetic data sets were needed to get to the current performance, and what the behaviour of the performance is as the number of datasets trained on are increased.

**Questions For Authors:**

In addition to responding to the comments regarding the weaknesses:
* Is it the case that the architecture is the main novelty of the paper?

**Relation To Broader Scientific Literature:**

* The paper is focused on ICL for performing Bayesian inference. The results show there is potential to leverage transformer-based architectures for performing Bayesian inference. What is less clear is how this work’s conclusions differ from the paper “Transformers can do Bayesian inference”.

**Theoretical Claims:**

* This does not seem to be applicable here. I do not believe any theoretical claims are made.

---

> ### Author Rebuttal · Authors · 2025-03-30
>
> Thank you for taking the time to read our manuscript and for providing detailed feedback.
>
> > Default hyperparameters are used for the VI approaches according to the appendix (no hyperparameter optimization)
>
> We would like to kindly point out that we investigate the role of the learning rate, which is a crucial hyperparameter for the VI methods, in appendix J of the manuscript. We furthermore decided to use standard “out-of-the-box” variational methods as baselines to allow for a direct with the in-context learner, which cannot even adapt its weights for each specific dataset.
> Furthermore, we include ablations in our paper (please see Appendix D) where we compare against strong diffusion-based baselines.
>
> > The key weakness that is preventing me from providing a higher score is in the identification of the novelty of the work in comparison to Müller et al.
>
> In their paper on PFNs, Müller et al. [1] focus exclusively on **univariate** posterior predictive distributions. Our approach, on the other hand, targets multivariate posteriors of latent variables. The key difference is thus in a scalar-valued distribution for PFNs versus a multivariate distribution in our approach. This is a fundamental difference comparable in scope to the difference between supervised learning (PFNs) and unsupervised learning (our method).
>
> This difference manifests itself in various aspects that include:
>
> - The tasks for which inference is learned in context:
>
>     - Müller et al: Regression with Bayesian neural networks, regression with Gaussian processes, and classification of hand-written digits.
>     - Ours: Full Bayesian inference regarding the latent variables of generalized linear models, factor analysis, and Gaussian mixture models.
> - The framework to learn the distributions
>     - Müller et al: Discretization of the distribution of interest and a cross-entropy loss.
>     - Ours: Flow matching.
>
> - The architecture
>     - Müller et al: A slightly modified standard transformer.
>     - Ours: A novel setup combining a  PFN-style transformer-encoder together with a diffusion transformer decoder.
>
> - The evaluation
>     - Müller et al: Predictive metrics and metrics for calibration
>     - Ours: Comparison of posterior samples from our method and the samples from HMC using measures for the similarity of distributions (MMD, C2ST, and W2 metrics)
>
> > how many synthetic datasets were needed to achieve the reported performances.
>
> Thank you for this question. We train on 37.5 million synthetic datasets, test on 30 million datasets, and validate on 7.5 million datasets. In terms of computational resources, each of our training runs takes at most three hours on an A100 graphics card. To put that into relation, for example, TabPFN version 1 is trained for 20 hours on 8 GPUs (Nvidia RTX 2080 Ti) and TabPFN version 2 even for 2 weeks on 8 Nvidia RTX 2080 Ti.
>
> Besides the existing detailed explanation in sections A and E of the appendix, we will include this information in the paper and thank the reviewer again for this suggestion.
>
> > The experiments on the OOD performance in the appendix are interesting and probably should be promoted to the main paper. I would suggest thinking of a new way of presenting these experiments though. Perhaps as a plot where the y-axis is the metric, and the x-axis is the KL-divergence between the training and test data distributions.
>
> Thank you for your comment. We agree that the OOD performance is an interesting aspect worth highlighting in the main body of our paper. Since we focus on controlled setups (GLM, FA, GMM) to enable comparisons to baselines from standard Bayesian inference (HMC and VI), limited OOD generalization is expected — a trade-off we confirm in Appendix E. We will make this clearer in the main paper.
>
> We also followed your great recommendation and created the [following plot](https://anonymous.4open.science/r/Extra-B820/plot_ood.pdf) which we will add to the new section in the main paper explaining the situation in terms of OOD performance.
>
> ## References
>
> [1] Müller, Samuel, et al. "Transformers Can Do Bayesian Inference." Eleventh International Conference on Learning Representations.
>
> [2] Lipman, Yaron, et al. "Flow Matching for Generative Modeling." The Eleventh International Conference on Learning Representations.

---

### Official Review · Reviewer_Y4eg · 2025-03-12

**Overall Recommendation:** 2

**Summary:**

The authors leverage transformers architecture to amortize Bayesian posterior estimation based on training data / observations fed in-context to the model. They conduct analysis on generalized linear models, factor analysis and mixture models and highlight that the proposed method discovers the true posterior distribution quite well, measured through classifier 2-sample test (C2ST), maximum mean discrepancy (MMD) and 2-Wasserstein ($\mathcal{W}_2$) metrics. Training such an amortized model is done through the lens of forward-KL minimization, which becomes tractable when amortized (i.e. the outer expectation w.r.t observations) and the parameterization of the approximate density is accomplished using flow matching. The paper shows that across a suite of synthetic and real world tasks, the proposed neural posterior estimation method stays competitive and outperforms reverse KL Variational inference when the metrics are computed through MCMC samples as proxy.

**Claims And Evidence:**

While the authors claim that they propose a general-purpose machinery for Bayesian posterior estimation, I will consider the contribution of the work as the application of existing approaches towards a suite of interesting and useful tasks, with a focus on studying its generalization capability to real world tasks. This is because the framework is quite similar to some of the existing works [1, 2] and is primarily the application of neural posterior estimation in simulation-based inference on the class of tasks described in the paper using flow matching and the transformer architecture.

In addition, the following claims made by the authors in their contributions section are not well supported -

- The authors claim that their method yields samples from the posterior distribution without parameter updates or parametric assumptions about the posterior. This is untrue as they do make parametric assumptions about the posterior, in the sense that the posterior is modeled through an ordinary differential equation with explicit parameters.

- The authors *do not provide a general framework to* analyze the circumstances that enable learning $P^{z|x}$ purely through samples from the joint. In particular, they do not provide any theory on when such a density can be reliably learned.

In addition, it is claimed in Section 3 that the proposed method does not suffer from overly or insufficiently flexible distribution assumptions as in VI. This is untrue, as one could also leverage continuous-time methods for reverse KL minimization, yielding the same family of distributions as the proposed work.

[1] Wildberger, Jonas, et al. "Flow matching for scalable simulation-based inference." Advances in Neural Information Processing Systems 36 (2023): 16837-16864.

[2] Mittal, Sarthak, et al. "Exploring exchangeable dataset amortization for bayesian posterior inference." ICML 2023 Workshop on Structured Probabilistic Inference {\&} Generative Modeling. 2023.

**Essential References Not Discussed:**

The authors should consider citing [1] which essentially looks at the problem of amortized posterior inference in known likelihood models like linear models, Bayesian neural networks and Gaussian Mixture Models through the lens of both forward and reverse KL methods using a Gaussian distribution or normalizing flows as the approximate density.

They should also cite [2] which, among other SBI methods, describes how forward KL can be used as a measure of divergence for training amortized posterior estimators.

[1] Mittal, Sarthak, et al. "Exploring exchangeable dataset amortization for bayesian posterior inference." ICML 2023 Workshop on Structured Probabilistic Inference {&} Generative Modeling. 2023.

[2] Radev, Stefan T., et al. "BayesFlow: Learning complex stochastic models with invertible neural networks." IEEE transactions on neural networks and learning systems 33.4 (2020): 1452-1466.

**Experimental Designs Or Analyses:**

Unfortunately, the descriptions of the experimental designs were unclear and / or potentially riddled with typos. Appendix A.1 in the write-up often confuses between $\mathbf{u}$, $\mathbf{z}$ and $\mathbf{x}$. For example, the authors claim that $\mathbf{x} := (\mathbf{z}, y)$ while also claiming that $\mathbf{z} = \mathbf{\beta}$, which should not be the case. Similarly, for Factor Analysis in Algorithm 3, the authors do not include $\mathbf{W_i}, \mathbf{\psi_i}, \mathbf{\mu_i}$ in the latent. Additionally, line 6 should be $\mathbf{z}_{i,j}$.

**Methods And Evaluation Criteria:**

Since the proposed method is aimed at posterior estimation, the benchmark datasets and evaluation criteria considered make sense and are relevant. However, there are a number of limitations in the evaluation procedure, which I detail below

- The paper does not contain any evaluation based on predictive performance. In particular, for supervised learning problems (i.e. GLM experiments) at least, it is imperative to also provide performance metrics (e.g. $l_2$ loss or the likelihood) and compare it to directly learning the linear method through gradient descent or estimating the posterior through MCMC / VI.

- Given that the work claims to use the data generating mechanism for TabPFN [1], it would be useful to get a comparative analysis in terms of predictive performance from TabPFN.

- For the VI-style baselines, the authors consider a non-amortized setup as baseline while using an amortized model as the proposed approach. An identical formulation can be framed from the reverse KL minimization perspective, and should be considered as one of the baselines. In particular, [2, 3] show that reverse KL often outperforms forward KL methods in terms of predictive metrics especially for high-dimensional tasks.

- Details regarding the dimensionality of tasks considered is missing and is extremely relevant. The authors should also provide some form of analysis into how their performance varies as a function of dimensionality of the task, at least for the GLM experiments. This is because with increasing dimensionality, it becomes harder and harder to maintain similarity to real-world tasks in the simulated data, and could lead to worsening performance on them.

[1] Hollmann, Noah, et al. "Tabpfn: A transformer that solves small tabular classification problems in a second." arXiv preprint arXiv:2207.01848 (2022).

[2] Mittal, Sarthak, et al. "In-Context Parametric Inference: Point or Distribution Estimators?." arXiv preprint arXiv:2502.11617 (2025).

[3] Mittal, Sarthak, et al. "Amortized In-Context Bayesian Posterior Estimation." arXiv preprint arXiv:2502.06601 (2025).

**Other Comments Or Suggestions:**

- The authors claim in line 146 that VI problems typically consider a simplified factorization of the variational density, which is not always true. The works cited (e.g. VAEs) perform this approximation as they consider latent variables corresponding to every observation (i.e. only local latent variables) while other works like Neural Processes [1,2] consider global latent variables which are shared across observations for a particular dataset. The authors should make it clear that they do not provide a more general assumption.

- The writing of the draft can be significantly improved. The authors often use the notation of $\mathbf{x}$ to either mean a dataset (e.g. line 187) or a meta-dataset (dataset of datasets; line 177).

- The authors claim that they do not make any assumptions about the latents $z \in \mathcal{Z}$ in footnote 2, regarding its decomposition into $z_i$. However, immediately afterwards in line 189 they consider a mapping $f_0$ which maps $\chi$ to $\mathcal{M}(\mathcal{Z})$ implying that $x_i$ maps to a density over $z$, which does imply that $z$ is local to each dataset.

- In light of the above two points, I would highly recommend the authors to provide a clearer write-up in their draft and make sure that the mathematical objects that they are working with are consistent. In particular, they can either describe everything in the general framework of $z \in \mathcal{Z}$ or work under the factorization $P^{z_i | x_i}$, but they should be consistent.

[1] Garnelo, Marta, et al. "Neural processes." arXiv preprint arXiv:1807.01622 (2018).

[2] Kim, Hyunjik, et al. "Attentive neural processes." arXiv preprint arXiv:1901.05761 (2019).

**Other Strengths And Weaknesses:**

**Strengths**

- The problem considered in this work is important and relevant and the authors conduct analysis on a variety of synthetic and real-world tasks which is useful.
- The work successfully shows that through posterior estimation metrics, amortization for posterior estimation uncovers the true posterior quite well.

**Weaknesses**

- The novelty of the work lies solely in their experimental results as the use of forward KL minimization, flow matching and amortization on transformers has been already studied considerably in related works.
- Benchmarking and evaluation of the method is limited as the authors do not provide predictive metrics, or compare to amortized VI methods.
- The work provides limited ablations; it could consider benchmarking the methods with different assumptions about the modeling distribution: flow matching, score-based diffusion, Gaussian approximation and discrete normalizing flows. Only two of the mentioned methods are tested.
- The writing needs more work; currently the mathematical formalism in the main paper and the data generation outline in the Appendix are not consistent.

**Questions For Authors:**

I found the following details to be incorrect, could the authors clarify if I am missing something?
- Equation 5 appears to be wrong. The objective for learning the vector field should be $z^{(1)} - \omega z^{(0)}$ (note the lack of division).
- Equation 6 needs to be pre-multiplied by $\frac{1}{N}$.
- Please use $\gamma_t(z^{(1)}, z^{(0)})$ (functional notation) as it is not a density, hence should not use the conditional operator.
- Line 300 introduces an additional notation of $\psi$ which is exactly same as the $\gamma$ considered before.
- Have the authors considered using a nonlinear neural network for computing C2ST? Could the authors do so and highlight if they get similar performance?

**Relation To Broader Scientific Literature:**

The key contributions of the work lie in their experimental setup, as prior works have leveraged transformers for amortization, forward KL minimization as the training signal for posterior estimation and flow matching as the parameterization of density. I think the work is well positioned and shows interesting experiments and outcomes, but needs more rigorous testing and benchmarking, as well as additional baselines and ablations.

**Theoretical Claims:**

The authors do not make any theoretical claims, except for Proposition 1 which is well known in literature and is the basis for neural posterior estimation methods within the framework of simulation-based inference. Please note that not making any theoretical claims is not a critique of the paper.

---

> ### Author Rebuttal · Authors · 2025-03-30
>
> We thank the reviewer for the valuable suggestions — we extended the experiments as proposed and now prominently highlight Mittal et al. as particularly relevant related work.
>
> > The key contributions of the work lie in their experimental setup [...].
>
> We fully agree that our core contribution lies in the experimental setup. However, we also (a) introduce a new architecture combining the PFN-type encoder and a modified version of diffusion transformers, (b) point out a sufficient condition that is central for many related amortized inference setups but has, to the best of our knowledge, not been made explicit in this form. Unlike your references [2] and [3], we (c) directly evaluate the quality of the posterior distribution.
>
> > Training such an amortized model is done through the lens of forward-KL minimization
>
> Please note that we do not use a KL-based objective, but train the in-context learner within the Flow Matching framework.
>
> > The authors claim that their method yields samples from the posterior distribution without parameter updates or parametric assumptions about the posterior.
>
> We use the term “nonparametric” in the sense that we make very weak assumptions about the underlying distribution of the data.
>
> In particular, we obtain a posterior distribution only implicitly defined through flow matching.
>
> > The authors do not provide a general framework to analyze the circumstances that enable learning purely through samples from the joint.
>
> We use the terminology “general framework” to refer to (a) our architectural framework in combination with Flow Matching, which arguably is a generic framework for learning distributions, and (b) Proposition (1) that, although a direct corollary of the law of total expectations, is to the best of our knowledge the first sufficient condition explicitly presented to describe  Neural Posterior Estimation [1], PFNs, Score Matching Posterior Estimation [2] and Flow Matching Posterior Estimation [3].
>
>
> >In addition, it is claimed in Section 3 that the proposed method does not suffer from overly or insufficiently flexible distribution assumptions as in VI. This is untrue, as one could also leverage continuous-time methods for reverse KL minimization, yielding the same family of distributions as the proposed work.
>
> A known benefit of in-context learners in the form of PFNs, and especially TabPFNs, is that they can flexibly adapt to the complexity of the presented data and thus do not require extensive hyperparameter tuning. This is what we mean by that claim. We will clarify this in the manuscript.
>
> > The paper does not contain any evaluation based on predictive performance.
>
> Please refer to our [new results](https://anonymous.4open.science/r/Extra-B820/Pred_Performance.pdf) regarding the predictive performance in the GLM scenarios. Those results show that forward-KL based VI and in particular MAP solutions perform strongly in terms of predictive performance, similar to results first discovered by your references [2] and [3].
>
> > [...] it would be useful to get a comparative analysis in terms of predictive performance from TabPFN.
>
> Please see the previous answer for our new results that also include a comparison to TabPFN.
>
> > [...] An identical formulation can be framed from the reverse KL minimization perspective, and should be considered as one of the baselines.
>
> We also conducted [comprehensive experiments](https://anonymous.4open.science/r/Extra-B820/FwdKL_Gaussian.pdf) investigating the performance of using a Gaussian approximation in the forward-KL setup.
>
> >Details regarding the dimensionality of tasks considered is missing and is extremely relevant.
>
> Please refer to our [new results](https://anonymous.4open.science/r/Extra-B820/Dim_Ablation.pdf). We will also rigorously adapt the claims and limitations section of our paper in light of the new results here.
>
> > Unfortunately, the descriptions of the experimental designs were unclear and / or potentially riddled with typos.
>
> We thank the reviewer for pointing out those typos! We fixed all the issues you mentioned.
>
> > The authors should consider citing [...]
> We thank the reviewer for pointing out those missing and clearly important references. We will include a discussion of those two papers and think that especially the work by Mittal et al. will nicely complement our findings while analyzing further interesting aspects of in-context learning.
> > The authors claim in line 146 that VI problems typically consider a simplified factorization of the variational density, which is not always true.
> Thank you for this remark. We will expand the section on related work on amortized inference to include your comment.
>
> ### Questions For Authors
> We thank the reviewer for the very helpful suggestions regarding different conventions in our notation and for pointing out typos. We will follow the reviewer's recommendations.

---

> > ### Comment · Reviewer_Y4eg · 2025-04-01
> >
> > Thanks to the authors for providing clarifications. Unfortunately, a number of my questions (re Questions For Authors) have not been addressed, especially regarding the correctness of Equation 5 and the comparison to C2ST using a nonlinear neural network.
> >
> > Additionally, the authors provide new results for ablating dimensions and posterior approximation but I cannot understand the results that they have in Table 1 Scenario 2 where it seems like posterior estimation improves on going from 5-dimensional tasks to 20. This seems counter-intuitive as it has been generally shown that posterior inference mechanisms get worse with increasing dimensionality of the problem.
> >
> > It is also surprising that the predictive performance of VI methods is low compared to that of the proposed methodology, and I suspect this is due to insufficient hyperparameter tuning for the VI methods. This finding is surprising because VI methods are supposed to be more mode-seeking, and should thus model a smaller volume around the mode as opposed to the full distribution. This means that they should not be very far away from the MAP-based solutions, however the tables suggest something different. Especially using the mean of DiagonalNormal for the prediction should be somewhat similar to the MAP predictions, if I understand things correctly? Could the authors clarify on why it is the case that a Gaussian VI is considerably worse than the proposed method for the specific case of predictive metrics?
> >
> > I would be happy to raise my score if the authors address my concerns above.

---

> > > ### Author Response · Authors · 2025-04-06
> > >
> > > We apologize for not answering some of your questions in our first answer in detail and would like to take this opportunity to address your questions below:
> > >
> > > > Equation 5 appears to be wrong.
> > >
> > > Thank you very much for spotting this error. We will fix it in the revised version of the manuscript. We would like to point out, however, that this error in the formula neither affects any of the other equations of the paper nor our implementation.
> > >
> > > > Equation 6 needs to be pre-multiplied by $\frac{1}{N}$.
> > >
> > > We agree that the term “empricial risk” is commonly used to refer to equation (6) divided by $N$, and will use the term “objective function” instead.
> > >
> > > > Please use (functional notation) as it is not a density [..]
> > >
> > > We initially decided to also use the conditional operator for a vector field to stay consistent with the notation introduced in [1]. However, we thank the reviewer for pointing out that this might cause confusion and will switch to the notation proposed by you.
> > >
> > > > Line 300 introduces an additional notation [...]
> > >
> > > This is indeed a mistake. The notation should be the same as before.
> > >
> > > > Have the authors considered using a nonlinear neural network for computing C2ST?
> > >
> > > Please refer to our extensive [new experimental results](https://anonymous.4open.science/r/Extra-B820/Ablation_CLF_for_C2ST.pdf) that confirm that using a neural network leads to overall very similar results.
> > >
> > > > Additionally, the authors provide new results for ablating dimensions and posterior approximation but I cannot understand the results that they have in Table 1 Scenario 2 [...]
> > >
> > > We agree with the reviewer that this is an interesting phenomenon that occurs in this specific scenario. However, it does not occur in any of the other 5 scenarios. To provide further evidence, we [ran the experiment](https://anonymous.4open.science/r/Extra-B820/Dim_Ablation.pdf) for the remaining GLM scenarios 1,4,6,7 where we also did not notice this behavior.
> > >
> > > While we also find that VI will typically become worse with increased dimensionality, the performance can potentially also improve in our case because of a better signal-to-noise ratio resulting from fixing the variance of the noise term while increasing the number of regression coefficients.
> > >
> > > > I suspect this is due to insufficient hyperparameter tuning for the VI methods
> > >
> > > Please note that the learning rate hyperparameter for the VI methods is chosen based on empirical results (Appendix L) and that we also do not tune the ICL method.
> > >
> > > > Especially using the mean of DiagonalNormal for the prediction should be somewhat similar to the MAP predictions [...].
> > >
> > > While the MAP estimate does consistently outperform other VI methods (Table 1), the predictive performance is in 13/14 cases within two standard errors of the best-performing VI method. For the dimensionality ablation, we find that the MAP performance is better compared to the VI methods with high dimensionalities.
> > >
> > > Overall, we think that this can be explained by two common issues with variational inference that can be particularly prominent in our case due to the relatively low number of data points per dataset ($N=50$) combined with a relatively high variance of the additive noise.
> > > - A misspecified variational family to optimize over: [2] show that this incurs a variational approximation error that only vanishes if the number of samples reaches infinity.
> > > - Overfitting due to more variational parameters than actual parameters. Even the considered DiagonalNormal approximationr has twice as many parameters as the Bayes-optimal model. This can lead to overfitting of VI methods [3].
> > >
> > > The MAP has neither of those problems.
> > >
> > > > Could the authors clarify on why it is the case that a Gaussian VI is considerably worse than the proposed method for the specific case of predictive metrics?
> > >
> > > We politely disagree with the statement that our results show that the Gaussian VI is **considerably** worse than the proposed method for the predictive metrics and do not think that we make this claim anywhere. For example, in our previous answer, we said that “[...] those results show that forward-KL based VI and in particular MAP solutions perform strongly in terms of predictive performance, similar to results first discovered by your references [2] and [3].”
> > >
> > > We think that the slightly better predictive performance in, for instance GLM scenario 4, is in line with the fact that HMC gives the best predictive performance and the ICL method yields samples often closest to HMC.
> > >
> > > We sincerely appreciate the reviewer's constructive feedback. As we have made every effort to respond thoroughly and substantively to all concerns, we would be grateful if the reviewer could kindly reappraise the initial score.
> > >
> > > ### References
> > > [1] Lipman, et al. "Flow Matching for Generative Modeling." ICLR. 2023
> > >
> > > [2] Wang, et al. "Variational Bayes under model misspecification." Neurips. 2019.
> > >
> > > [3] Cremer, et al.. "Inference suboptimality in variational autoencoders." ICML. 2018.

---

### Official Review · Reviewer_PknZ · 2025-03-25

**Overall Recommendation:** 3

**Summary:**

The authors present an approach for (approximate) Bayesian inference, based on in-context learning with a combined transformer/flow model. The approach relies on access to the generative model, and generates data to learn the inverse model. The method is reasonable and performs well relative to baselines.

**Claims And Evidence:**

Overall, the claims are supported, although the baselines and experimental evaluation are relatively weak (discussed below).

**Essential References Not Discussed:**

The authors may find "General-purpose in-context learning by meta-learning transformers" interesting as another example of ICL on (partially) synthetically generated data, but citing this paper isn't critical.

**Experimental Designs Or Analyses:**

The major weakness of this paper is the experimental evaluation:
- The paper is missing an application that motivates a reader, and it is currently not clear that one really exists. I could imagine that this approach would be useful for situations in which some form of parameter identification is desirable and must be done repeatedly, but the authors really must add an experiment with a real application.
- The authors only compare to simple VI methods and Laplace approximation. The use of LA for this problem seems somewhat strange, it is not clear to me why this would be a sensible approach to this problem. There is an existing literature on Bayesian methods for inverse problems that would provide better methods, and the authors could also compare to other amortized variational methods beyond the IAF. This is partially addressed in the diffusion objective ablation, which should probably be presented in the paper body.
- While there is some discussion of robustness to OOD in appendix H, the model still feels quite sensitive to the data generation process and a more thorough characterization of robustness to model misspecification is likely necessary. This is especially pertinent for real world data.

**Methods And Evaluation Criteria:**

The authors should better justify applications in which the this degree of amortized inference is necessary/useful. There are many inverse problems, for which simulation-based data generation is done, that could perhaps provide better motivation than what is currently provided.

**Other Comments Or Suggestions:**

None

**Other Strengths And Weaknesses:**

Overall I found the paper to be fairly well presented, although the authors could mention what their approach is (transformer encoder + flow matching) earlier in the paper for people who are skimming the paper.

**Questions For Authors:**

Feel free to respond to the weaknesses I stated, but I have no other questions.

**Relation To Broader Scientific Literature:**

The work exists within the literature of approximate/amortized Bayesian inference. It also overlaps with the literature on Bayesian inverse problems. The authors discuss approximate methods in Bayesian inference, and discuss TabPFN in detail. The authors could provide better motivation for why their approach may be useful than they currently do.

**Theoretical Claims:**

The mathematical results appear to be correct, and are primarily direct application of previous results.

---

> ### Author Rebuttal · Authors · 2025-03-30
>
> Thank you for taking the time to read our manuscript and for providing detailed feedback.
>
> > The paper is missing an application that motivates a reader, and it is currently not clear that one really exists.
>
> We would like to point out that the question addressed in our paper is “can transformers learn full Bayesian inference in context?”, which we see and treat as a fundamentally conceptual question. Our work thus stands in line with papers such as [1-2] that investigate different aspects of in-context learning from a conceptual perspective. However, unlike the aforementioned papers [1-2], we also present results on real-world data. We see this as the next, substantially more difficult step to benchmark for the conceptual capabilities of the proposed in-context learning approach.
>
> However, we think that learning full Bayesian inference in context has **conceptually** great promise to alleviate important issues with traditional Bayesian inference, including
>
> (a) slow inference time of methods like MCMC. Once pre-trained, an in-context learner can provide samples quickly.
>
> (b) inferior predictive performance compared to non-Bayesian approaches: TabPFN [3] and related approaches show that carefully designing a realistic synthetic distribution to train an in-context can lead to Bayesian methods with excellent predictive performance.
>
> > The authors only compare to simple VI methods and Laplace approximation.
>
> The main concern of our benchmarks is to demonstrate that transformers are capable of in-context learning full Bayesian inference akin to other recent in-context learning works analyzing whether transformer can learn optimization routines or decision boundaries of other methods in-context, e.g. [1-2]. We thus compare our method against popular and commonly used methods in variational inference as well as HMC for the latent variable models we analyze.
>
> Please also note that the Laplace approximation, while we fully agree is certainly an outdated method for the considered problems, serves as a simple baseline.
>
> > There is an existing literature on Bayesian methods for inverse problems that would provide better methods
>
> First, we would like to point out that approaches that also operate in the flow-matching framework are indeed state-of-the-art methods for Bayesian inference in inverse problems [5]. We further compare against the recently proposed score-matching-posterior-estimation [3] via our ablations in appendix E. Our [new results](https://anonymous.4open.science/r/Extra-B820/FwdKL_Gaussian.pdf) are a comparison further method that can also be used for simulation-based inference.
>
> Please note that our goal is not to propose a new method for inverse Bayesian problems but to show that in-context learning can be effective, even compared to well-established methods for Bayesian inference that do not operate in context, which present an arguably even stronger baseline.
>
> >While there is some discussion of robustness to OOD in appendix H, the model still feels quite sensitive to the data generation process [...]
>
> We fully agree that in our setup, the data-generating process we use to train the in-context learner determines what datasets the in-context learner can provide meaningful inference on. Please note, however, that this is expected as we consider controlled and simple setups (GLM, FA, and GMM scenarios) to perform full Bayesian inference. We think that having a somewhat artificial setup of well-established latent variable models where strong traditional baselines exist (HMC, VI) is crucial for exploring whether transformers can do full Bayesian inference in context.
>
> As you correctly point out, however, this comes at the cost of inherent limitations in terms of OOD performance, which we confirm in our ablation study in appendix E. In principle, however, our in-context learner could be applied to compley, more realistic setups similar to what TabPFN does; but then we would lose the ability to do meaningful evaluations.
>
> > The authors may find "General-purpose in-context learning by meta-learning transformers" interesting [...]
>
> Thank you for pointing out this relevant and interesting reference. We will discuss its relation to our work in the revised related work section.
>
> > The authors could mention what their approach is [...] earlier in the paper
>
> Thank you for this very helpful remark! We will add a short explanation to the introduction.
>
> ## References
>
> [1] Garg et al. What can transformers learn in-context? NeurIPS 2022.
>
> [2] Panwar, Ahuja & Goyal. In-Context Learning through the Bayesian Prism. ICLR 2024.
>
> [3] Gloeckler, Manuel, et al. All-in-one simulation-based inference. ICML 2024.
>
> [3] Hollmann et al. Accurate predictions on small data with a tabular foundation model. Nature, 2025.
>
> [3] Ahuja et al. In-Context Learning under Distribution Shifts. Workshop on Efficient Systems for Foundation Models @ ICML 2023.

---

### Decision · Program_Chairs · 2025-05-01

**Decision:**

Accept (poster)

**Comment:**

This paper presents a method for using in-context learning in transformer based models to perform full Bayesian inference.  The authors use the transformer to learn a conditional continuous normalizing flow that parameterizes a vector field of an ordinary differential equation (ODE).  An ODE solver can then be used to sample from some simple base distribution to the target posterior of interest.  The paper received borderline but positive reviews (3, 2, 3, 3).  The reviewers found the paper well-written, well structured and mathematically sound.  They found the approach novel and the experimental results presented strong.  In terms of weaknesses, the reviewers asked for stronger baselines, better metrics and were concerned about scalability.  There was a bit of a sense that reviewers found it interesting that one could do Bayesian inference this way, but they didn't necessarily get why one would do Bayesian inference this way.  Perhaps the authors could have motivated the approach a bit better and given some insight into how this would be used in practice over performing standard Bayesian inference (e.g. MCMC) using the actual model.   There was some concern that the approach isn't scalable, which adds to the question of why one would want to use a giant transformer-based model to approximate a simpler model in-context.  Nevertheless, the paper is insightful and technically interesting and seems like a novel and fun way to co-opt in-context learning to do something more sophisticated.